# Transcriptome Profile Alteration with Cadmium Selenide/Zinc Sulfide Quantum Dots in *Saccharomyces cerevisiae*

**DOI:** 10.3390/biom9110653

**Published:** 2019-10-25

**Authors:** Cullen Horstmann, Daniel S Kim, Chelsea Campbell, Kyoungtae Kim

**Affiliations:** 1Department of Biology, Missouri State University, 901 S National, Springfield, MO 65897, USA; Horstmann95@live.missouristate.edu (C.H.); cnc8223@gmail.com (C.C.); 2Kickapoo High School, 3710 South Jefferson Ave, Springfield, MO 65807, USA; Dandude1244@gmail.com

**Keywords:** QDs, toxicity, yeast, CdSe/ZnS, RNA-seq, gene expression

## Abstract

Quantum Dots (QDs) are becoming more prevalent in products used in our daily lives, such as TVs and laptops, due to their unique and tunable optical properties. The possibility of using QDs as fluorescent probes in applications, such as medical imaging, has been a topic of interest for some time, but their potential toxicity and long-term effects on the environment are not well understood. In the present study, we investigated the effects of yellow CdSe/ZnS-QDs on *Saccharomyces cerevisiae*. We utilized growth assays, RNA-seq, reactive oxygen species (ROS) detection assays, and cell wall stability experiments to investigate the potential toxic effects of CdSe/ZnS-QDs. We found CdSe/ZnS-QDs had no negative effects on cell viability; however, cell wall-compromised cells showed more sensitivity in the presence of 10 µg/mL CdSe/ZnS-QDs compared to non-treated cells. In CdSe/ZnS-treated and non-treated cells, no significant change in superoxide was detected, but according to our transcriptomic analysis, thousands of genes in CdSe/ZnS-treated cells became differentially expressed. Four significantly differentiated genes found, including *FAF1*, *SDA1*, *DAN1*, and *TIR1*, were validated by consistent results with RT-qPCR assays. Our transcriptome analysis led us to conclude that exposure of CdSe/ZnS-QDs on yeast significantly affected genes implicated in multiple cellular processes.

## 1. Introduction

Quantum Dots (QDs) are extremely small colloidal semiconductor nanoparticles (NPs) typically 1–10 nanometers in diameter. They are a diverse group of nanomaterials (NMs) that are classified based on physical properties, such as their size, charge, shape, and the chemical composition of their core and shell [1]. These materials (typically Cd-QDs) are an attractive topic in research due to their unusual optical characteristics, mainly, their photo-stability, narrow-tunable emissions, and broad excitation ranges [2,3]. They have become widely incorporated in electronics, agriculture, and textile production, but they are mostly sought out for their biomedical applications (cellular and protein labels, real-time trackers, fluorescence resonance energy transfer (FRET) sensors, etc.) and used as a smart drug delivery system (SDDS) for treating cancer [1,3,4,5,6,7,8]. They make excellent fluorescent probes because their optical properties are size-dependent and are easily manipulated. In addition, they resist photo-bleaching and produce a greater brightness than conventional organic dyes [3,4]. QDs make a prime candidate for use as nanocarriers in SDDSs because, unlike other nanocarriers, they can simultaneously visualize tumors in addition to delivering a drug to its target [5]. Although QDs are diverse and utilized in numerous applications, there is an increasing concern on their leakage and long-term effects on the environment and human health [1,6].

Previously published works on QDs present conflicting results regarding cytotoxicity, but most articles that investigate their effects, *in vitro* and *in vivo*, seem to agree that their physiochemical properties, such as size, charge, composition, and concentration, are responsible for their toxicity [2,6]. Herein lies the challenge of studying QD toxicity. Their high possible combinations of physiochemical properties result in a broader spectrum of toxic effects. It’s been reported that small 2.2 nm CdTe-QDs localize in the nuclear compartment, and the same QDs at 5.2 nm localize in the cytosol [6]. Negatively charged zwitterionic QDs with functionalized surfaces reduce mitochondrial activity by up to 25%, and cellular impedance is reported due to receptor-independent entry through the membrane [4]. However, positively charged CdSe/ZnS-QDs have been found to be less toxic [3]. CdTe-QDs exhibit a dose-dependent cytotoxic effect on cell viability, membrane integrity, metabolic activity, mitochondria integrity, and chromatin quality in an array of cells (HeLa, MCF-7, and NIH/3T3) [9]. The potential number of interactions between QDs and biological components are high, leaving essential questions regarding their toxicity unknown [3].

Previous studies have revealed that long-term exposure of 20 nM CdSe/ZnS-QDs, coated with polyethylene glycol (PEG), amines, or carboxylates, to the eye results in decreased cell viability [10]. Another study found that injecting 0.5 nM CdSe or CdSe/ZnS-QDs into the hippocampal area in rats impairs synaptic activity. This was thought to be induced by increasing calcium levels and Cd^2+^ ions that lead to defects in neuro-secretion [11,12]. In addition, they have been found to accumulate in the liver and kidneys in rats and could release Cd^2+^ ions in the body of the individual being exposed [13]. On the cellular level, CdSe/ZnS-QDs can enter the cell through the plasma membrane and have been found to inhibit viability in macrophage [14,15], human keratinocyte HaCaT [16], and human dermal fibroblast cell lines exposed to 15 nm QDs at concentrations of 30–60 nM [17]. Previous studies have found that the degradation of Cd-based QDs releases harmful Cd^2+^ ions that indirectly increase ROS levels (typically H_2_O_2_, ·O_2_^−^, and ·OH), capable of damaging proteins and membrane lipids, inhibiting DNA repair, disrupting cellular signaling, and causing apoptosis [18,19,20,21]. In addition, the precipitation of QD aggregates on the surface of the cell may impair the integrity of the cell wall [19]. In yeast, CdTe-QDs have been shown to exhibit cytotoxicity at concentrations as low as 80.81 and 17.07 nmol/L for green and orange emitting QDs, respectively [22]. Xiaole Han et al. revealed that QDs as small as 4.1 to 5.8 nm could be internalized in *Saccharomyces cerevisiae* and induce cytotoxicity through cell wall breakage and cytoplasm blebbing. Nevertheless, the details on what molecular mechanisms contribute to Cd-based QD toxicity are still poorly understood. To this end, our RNA-seq revealed more in-depth information on the processes and mechanisms that might be responsible for Cd-QD-induced toxicity.

Research papers investigating QD toxicity typically look at mechanisms, such as cell viability and induction of reactive oxygen species [3]. Gene expression assays are not common in studies on Cd-QD-induced toxicity. Using high-throughput quantitative reverse transcript polymerase chain reaction (qRT-PCR) assays, recent studies reported differentially expressed genes affected by CdSe/ZnS-QD toxicity, including genes involved in cellular stress and toxicity, DNA damage and repair, mitochondrial function, proliferation, and ovarian function in vivo [3,23,24]. RNA-seq has become the standard for assessing entire genomes and identifying differentially expressed genes (DEGs), and Simon et al. (2013) utilized this process to investigate the effects on transcriptomic profiles of green algae exposed to CdTe/CdS-QDs. They reported via a gene ontology (GO) analysis that DEGs were involved in oxidative-stress, redox potential, protein folding, and chaperone activity pathways in the Cd-treated cells [25]. Hosiner et al. (2014) conducted a microarray experiment on *Saccharomyces cerevisiae* that was exposed to several different metal salts to observe the effects of different metal ions on yeast’s transcriptional profile [26]. Their study revealed anti-oxidative genes (*GRX2*) and redox homeostasis genes (*TRR1*, *TRR2*, and *TRX2*) to be upregulated in response to CdCl_2_ exposure, due to potential release of Cd^2+^ ions [26]. Interestingly, cadmium is not redox-active, which means it cannot generate ROS directly, yet, Cd-induced ROS is a commonly observed response [27]. They pointed out that metals, such as As^3+^, Cd^2+^, and Hg^2+^, had an affinity toward thiol groups (-SH), which play disparate roles in the function of enzymes, transcription factors, and membrane proteins [26]. A more recent study conducted in 2016 employed RNA-seq to assess differences in gene expression when exposed to 320 μM CdSO_4_ in *Saccharomyces cerevisiae* [27]. They sorted DEGs into functional classes and found upregulated DEGs belonged to classes, such as transcription factors involved in GSH metabolism, proteins of cellular response to oxidative stress and regulation, enzymes of carbohydrate metabolism, proteins with antioxidant properties, mitochondrion related proteins, peroxisome, and other regulator/transcription factors, while downregulated DEGs belonged to a class of normal expression genes under anaerobic condition (*DAN1*, *AAC3*, *ANB1*, and *YER188W*) and heme biosynthesis key genes (*HEM3* and *HEM13*). In addition, their study revealed that CdSO_4_ decreased the mitochondrial membrane potential by over 52% and significantly increased ROS levels [21]. However, little is still understood on the transcriptional profiles of *Saccharomyces cerevisiae* when treated with a non-ionizing, Cd-based QD (CdSe/ZnS) that possesses a ZnS shell meant to prevent any harmful Cd^2+^ from leaking out of the CdSe core.

Though CdSe/ZnS-QDs have been used in many ways and conjugated or coated with various molecules and exposed to a variety of organisms, their impact on cellular environments and gene expression is not well understood and raises concerns about their potential toxicity, despite their “safe” core/shell structure. Chibli et al. found other “safer” core/shell QDs, such as InP/ZnS-QDs, generating ROS despite their ZnS shell [28]. Their study attributed the generation of ROS in NIH3T3 fibroblasts, KB cells, B16 murine melanoma cells, and MDA-MB-231 breast adenocarcinoma cells to the poor coordination strength between the InP core and ZnS shell, resulting in an unstable core/shell relationship that left the InP core exposed in some areas [28]. With the addition of a second ZnS shell around InP/ZnS-QDs, exposed sections of the core were contained, and a decrease in ROS generation was observed [28]. It was noted that a CdSe core and ZnS shell had a better coordination strength, resulting in a stable core/shell structure with minimum CdSe core exposure [28]. Due to these interesting results, it is unlikely that CdSe/ZnS-QDs require a second ZnS shell for their safe use. In addition, our CdSe/ZnS-QDs were synthesized with a carboxylic acid stabilizing ligand that is capped on the surface on the ZnS shell. Capping ligands are often used to prevent QDs from aggregating, and some may play a major role in the uptake of QDs into cells and where they are localized. Kunstman and associates (2018) bio-conjugated CdTeS/ZnS-QDs with a galactose ligand and successfully achieved fluorescent imaging of yeast cells [29]. They revealed that CdTeS/ZnS-QDs capped with galactose ligands accumulated in the membrane of yeast cells, while CdTeS/ZnS-QDs with unmodified surfaces failed to accumulate in the membrane or enter the yeast cells, suggesting that specific associations between the ligands and cell surface may play a role in the entry of Cd-QDs into yeast cells [29]. Due to CdSe/ZnS-QDs seemingly more stable structure than other core/shell QDs, their effects on cell physiology have remained elusive and require further explanation. The present study utilized deep sequencing technologies like RNA-seq to assess to what extent CdSe/ZnS-QDs affect the transcriptome profile of *Saccharomyces cerevisiae* with great precision.

## 2. Materials and Methods

### 2.1. CdSe/ZnS Quantum Dots

CdSe/ZnS-QDs (catalog number CZW-Y) with an emission color of yellow and a carboxylic acid stabilizing ligand with <1% organic impurities (not including ligands), suspended in water (1000 μg/mL), were obtained from NN-Labs (Fayetteville, AR, USA). The ZnS shell around the CdSe-QD core protects and stabilizes the QD’s unique optical properties while maintaining the same absorption (estimated 550–600 nm) and emission (570–585 nm) properties of the core. The resulting core/shell nanocrystals demonstrate brighter yellow fluorescence with greater stability and process-ability and increase control over the QD’s surface chemistry. NN-Labs did not provide the size of their yellow CdSe/ZnS-QDs, and the size was not available on the NN-Labs website. Baig et al. and associates, via transmission electron microscopy (TEM), determined the sizes of CdSe/ZnS-QDs with emission colors of green, yellow, and red to be 3.0, 4.1, and 5.5 nm, respectively [30]. These results led us to assume the size of our yellow CdSe/ZnS-QDs to be approximately 4.1 nm.

20 nm PELCO^®^NanoXact™ silver nanoparticle (AgNPs), suspended in a concentration of 20 µg/mL in 2 mM sodium citrate solution (pH 7.6), was obtained from Ted Pella, Inc (Redding, CA, USA. The average diameter of the spheroidal nanoparticles (NPs) was 20 ± 2.9 nm, measured by JOEL 1010 transmission electron microscope (for more chemical and physical information, visit: www.nanocomposix.com). This sample of NPs showed the absorption band at 393 nm (www.nanocomposix.com).

### 2.2. Growth Assay with Exposure to CdSe/ZnS Quantum Dots

Wild type *Saccharomyces cerevisiae* cells (S288C) were purchased from ATCC (American Type Culture Collection, Manassas, VA, USA) and cultured in synthetic defined glucose (SD-Glucose) media overnight in a shaking incubator (INFORS HT Minitron) at 30 °C. The optical density (OD) was recorded at 600 nm with a BioMate^TM^ 3S spectrophotometer (Thermo Scientific, Waltham, MA, USA). The cells were cultured for 16–18 h in the shaking incubator to a minimum concentration of 1 × 10^7^ cells/mL. After confirming the OD was adequate, the cells were inoculated into a 2x SD-Glucose media stock to an OD of 0.1. The newly made stock of cells was plated on a 96-well culture plate following the plating of CdSe/ZnS-QDs at concentrations of 0, 0.8, 1.6, 3.15, 6.25, 12.5, 25, 50, and 100 μg/mL in a triplicate manner. Upon completion of plating of the QDs and cells, the culture plate was inserted into an ELx808^TM^ absorbance microplate reader (Biotek, Winooski, VT, USA) and grown while shaken fast for 24 h at 30 °C and simultaneously recording the OD every 20 min at a wavelength of 594 nm. Blank well ODs were subtracted from the varying QD test concentration wells and averaged to create growth curves that represent each of the eight test concentrations, and then were compared to the growth curves created from the averaged positive control wells. The log section of the growth curves was used to find doubling times for each QD concentration investigated. The entire growth curve assay was repeated three times.

### 2.3. Total RNA Extraction

*Saccharomyces cerevisiae* (S288C) cells were cultured in SD-Glucose media to mid-log phase. The OD was determined at a wavelength of 600 nm and fell in the mid-log phase OD range 0.3–0.6. From the mid-log phase, some samples were treated with CdSe/ZnS-QDs. The cells were incubated at 220× *g* at 30 °C for six hours in a shaking incubator and brought to a concentration of 10 μg/mL CdSe/ZnS-QDs. A control sample containing no CdSe/ZnS-QDs was brought to 10 μg/mL. The RiboPure^TM^ yeast RNA extraction kit (Thermo Fisher Scientific) was used to perform a total RNA extraction on three control and three CdSe/ZnS-QD-treated samples. Once extracted, the RNA was quantified by measuring the OD at a wavelength of 280 nm with a Qubit 3.0 fluorometer (Thermo Fisher Scientific, Waltham, MA, USA). Final RNA concentrations fell in the acceptable range of 960–1200 ng/μL.

### 2.4. mRNA Isolation and cDNA Synthesis 

TruSeq^®^ stranded mRNA LT sample preparation kit (Illumina, San Diego, CA, USA) was used to isolate mRNA from the total RNA collected in the previous step. The kit provided superScript II reverse transcriptase that was used to synthesize cDNA strands from the newly purified mRNA. Using unique adaptors ligated to each cDNA sample, the samples could be distinguished when sequenced and then amplified for 15 cycles in a T100TM thermal cycler (BIO-RAD, Hercules, CA, USA). Next, the cDNA from the treated and untreated samples were suspended in 30 μL of resuspension buffer at a concentration of approximately 50 ng/μL. Illumina Hiseq 2500 sequencing system (Kansas Medical Genome Center, Kansas City, KS, USA) was used to sequence the three treated and untreated samples of cDNA.

### 2.5. Analysis of Sequencing Data

Data from cDNA sequencing were analyzed using Galaxy, a website platform for analyzing sequenced data (www.usegalaxy.org). The data obtained from the Kansas Medical Genome Center was uploaded to the Galaxy server, where sequences that were separated, when sent to us, were concatenated back together so the full reads could be analyzed. A quality check was carried out on each file of sequence data to check the quality of the reads and ensure good samples and that the data is interpreted correctly. The files were then re-formatted into Sanger, which is necessary for the steps to follow. To achieve high fidelity, the files were trimmed based on quality, and bases with a quality score below 20 were removed from the reads. For eliminating the bias of primers and to ensure the removal of adapters, 12 bases were trimmed from the 5′ end of the reads. The remaining reads were then filtered to remove any reads less than 80 base pairs. Next, reads were aligned to the wild type *Saccharomyces cerevisiae* reference genome (S288C) obtained from the Saccharomyces genome database (SGD) (YeastGenome.org) with Tophat in Galaxy. With Cufflink, the transcriptome was assembled using the reference annotation by comparing the reads to the reference genome. Lastly, using Cuffdiff, the aligned sequence expression rates were compared between sample conditions, creating a list of differentially expressed genes (DEGs). When the final differential gene data was obtained, genes with a *q*-value greater than 0.05 were not included in the final list of genes analyzed. The remaining genes were grouped based on correlating gene ontology (GO) terms obtained from GOrilla.

### 2.6. Quantitative Reverse Transcription PCR (RT-qPCR)

cDNA was synthesized from 1000 ng of total RNA from samples extracted from three control and three CdSe/ZnS-QD-treated cell cultures with the Verso cDNA conversion kit (Thermo Fisher Scientific). The resulting cDNA was quantified using a Qubit 3.0 fluorometer. After designing DNA primers for target genes (*FAF1*, *SDA1*, *DAN1*, *TIR1*, and *ALG9*), we performed a primer efficiency test to validate their efficacy in our cDNA samples to ensure they were adequate for RT-qPCR experiments. The genes listed above were selected because *FAF1* and *SDA1* were found to be upregulated and *DAN1* and *TIR1* to be downregulated in CdSe/ZnS-QD-treated samples. *ALG9* was chosen because its expression did not statistically change when exposed to CdSe/ZnS-QDs. The primer efficiency test consisted of serially diluted cDNA samples by a factor of 5. Each sample was then amplified with PCR using the GoTaq qPCR kit (Promega, Madison, WI, USA). With MxPro^®^ software (Agilent, Santa Clara, CA, USA), we calculated the primer efficiency and R-squared values. Primer efficiency values from all five primers fell within the range of 1.69 to 1.74, and the R-squared values fell within 0.99 and 1.00, indicating the sample dilutions and experiment preparation were precise. With good primer efficiency, we then used 60 ng of cDNA from our control and QD-treated samples to amplify our five target genes, using the GoTaq qPCR master mix protocol (Promega). Every target gene reaction amplified contained primers, GoTaq master mix, nuclease-free water, and cDNA. In addition, a non-treated control reaction containing no cDNA was prepared. Each well was mixed thoroughly by pipetting up and down several times before the plate was centrifuged for one minute to bring all of the reaction to the bottom. The plate was then moved to a pre-heated MX3005p instrument for PCR amplification. The Pfaffl method was utilized to find the fold-change in each target gene expression by comparing it to the expression of a housekeeping gene, *ALG9*. The target gene’s relative expression ratio was calculated based on their E (RT-qPCR efficiencies) and CP (crossing point) deviation compared to the control (*ALG9*), and the expression of the target genes was compared to the expression of the gene *ALG9*.

### 2.7. Measurement of ROS

The ROS measurement was performed by culturing yeast cells to an OD of 0.1 in Falcon tubes and separated into three groups: the non-treated control, 5 µg/mL AgNP-treated cells, and 20 µg/mL CdSe/ZnS-treated cells. Each group was created in a triplicate manner and incubated and shook for 6 h at 30 °C. After the incubation period, the contents of each Falcon tube were treated with a concentration of 0.5 mg/mL of dihydroethidium (DHE) and incubated for two hours in the dark while shaking at 30 °C. Next, 1X PBS buffer was added to each tube, and the fluorescent intensities of the oxidized DHE were measured utilizing an Attune NxT acoustic focusing cytometer (Life Technologies, Carlsbad, CA, USA), with the filter set at an excitation wavelength of 500 nm and an emission wavelength of 600 nm. The fluorescent intensities of the treated and non-treated cells were averaged and compared to one another, followed by a student’s *t*-test, where *p*-values higher than 0.05 were not considered statistically significant.

### 2.8. Nanoparticles’ Effects on Cells Lacking Cell Walls

The Zymolase assay was performed by culturing yeast cells in 3 mL of SD-Glucose overnight in a shaking incubator at 30 °C. The cell culture was then centrifuged at 2000× *g* for ten minutes, and the resulting cell pellet was re-suspended with 2X TE buffer to an OD of 1.0. The suspended cells were applied to a 96-well plate in a quadruplicate manner. To test the effects of CdSe/ZnS, the following samples were prepared: non-treated controls (cells with TE buffer), 10 µg/mL CdSe/ZnS-treated cells, and 20 µg/mL CdSe/ZnS-treated cells. To test the effects of AgNPs, samples prepared in a 96-well plate consisted of non-treated controls and 2.5 µg/mL and 5 µg/mL AgNP-treated cells. Cell walls were degraded by the introduction of Zymolase at a concentration of 0.5 µg/mL and incubated in an ELx808^TM^ absorbance microplate reader (Biotek, Winooski, VT) for four hours at 30 °C. During the incubation period, the optical density (594 nm) was measured in 30 min intervals, and each sample was independently tested without the use of Zymolase to serve as a control. The resulting changes in optical densities were recorded and averaged for each sample before plotting into a line graph.

### 2.9. Statistical Analysis

All experiments were performed, at least, in a triplicate manner, meaning each line and bar in a graph represents the average of three replicates. Additionally, all standard deviations were represented in each bar graph with error bars. The ANOVA: Single Factor test is a form of statistical analysis that reveals significant differences in an overall group. It was performed by utilizing the data analysis toolbox in Microsoft Excel and selecting the “ANOVA: Single Factor” option and then inputting the required data. If the *F*-value is lower than the F-critical value when the analysis is finished, then it is considered that there is no significant difference. Regardless of any significant differences, we performed a student’s *t*-test by utilizing the data analysis toolbox in Microsoft Excel, selecting “*t*-Test: Two-Sample Assuming Unequal Variances” and inputting the required data. When evaluating the two-tail *p*-values, a *p*-value of less than 0.05 was considered statistically significant, while a value between 0.05 and 0.1 represented minimal statistical significance.

## 3. Results

### 3.1. CdSe/ZnS-QDs Negatively Affect Yeast Growth

To investigate the effects of CdSe/ZnS-QDs on yeast growth, we utilized AgNPs as a positive control, as they have been shown to cause growth defects in yeast cells by Horstmann and coworkers [31]. As expected, the treatments of AgNPs obtained results consistent with the findings of Horstmann and coworkers, where the concentrations of 5 µg/mL and 10 µg/mL of AgNPs showed significant growth rate reduction when compared to non-treated controls (Figure 1A,B). In contrast, CdSe/ZnS-QDs did not show any effects on yeast growth when compared to the non-treated controls, even when treated with 100 µg/mL of CdSe/ZnS (Figure 1A,B). In order to more clearly define the differences of cell growth in the steady-state, we analyzed the last optical density value (endpoint OD_600 nm_ at 24 h) for both the AgNPs and CdSe/ZnS-QD-treated cells. In the CdSe/ZnS, there was no significant difference in endpoint OD values compared to the non-treated control according to the ANOVA test (Figure 1C,D). Similarly, the endpoint OD of AgNPs showed no significant difference from the non-treated control, via an ANOVA test (Figure 1C,D). However, when performing a student *t*-test on ODs exposed to both AgNP and CdSe/ZnS-QDs, the results revealed the endpoint ODs were statistically different from those of the non-treated control in cells treated with 5 µg/mL AgNPs (20 nm) and 6.25 µg/mL CdSe/ZnS-QDs (estimated 4.1 nm). During the exponential growth period, we analyzed the difference in doubling times between the non-treated controls and the treated cells. For the CdSe/ZnS-treated cells, the student *t*-test revealed that the average time spent in exponential growth was not significantly different from the non-treated control. As for the AgNP-treated cells, the average doubling time was significantly different from the non-treated control for most of the concentrations, according to the student *t*-test. Before the cells grew exponentially, the cells treated with either AgNPs or CdSe/ZnS-QDs showed, based on the student *t*-test, no significant difference from the non-treated controls. Taken together, based on the results obtained from the endpoint ODs, doubling times, and lag times, CdSe/ZnS did not have any negative growth effects on yeast cell growth, whereas the AgNPs showed adverse effects on yeast cell growth.

### 3.2. cDNA Sequencing Reveals Up- and Downregulated Genes with CdSe/ZnS-QDs

Instead of relying on the limited simple methods of proliferation, organelle integrity, or metabolic assays to gain insight on how CdSe/ZnS QDs interact with fungal cells, we decided to look into differential gene expression profiles to examine a broader range of cellular processes being affected. We determined the transcriptomic response in *S. cerevisiae* exposed to 10 μg/mL CdSe/ZnS-QDs by performing an RNA-seq that produced gene expression profiles for both the control and CdSe/ZnS-treated cells. Briefly, the control and CdSe/ZnS exposed cells were subjected to a total RNA extraction. Then, the mRNA was isolated from the total RNA, followed by a cDNA conversion step. Both control and QD-treated samples were tested in triplicate, and the newly synthesized cDNA libraries were sequenced with a next-gen DNA sequencer (Illumina^®^, San Diego, CA, USA) that produced sequenced datasets for each replicate. Each cDNA dataset had to be uploaded to a computational data analysis platform (usegalaxy.org) for processing, and all control and CdSe/ZnS-treated replicates were concatenated, leaving the first dataset composed of the three control samples and the second of the QD-treated samples. Every cDNA fragment underwent a quality check (FastQC) and quality trimming (FASTQ Quality Trimmer) before being mapped to the reference genome (S288C). An average of 19,619,921 accepted reads was gathered from the control groups and 19,205,868 from the CdSe/ZnS-treated groups. Of these quality reads, an average of 91.7% and 93.6% of the total reads mapped to the reference genome, and an average of 9.9% and 10% of the mapped reads had multiple alignments in control and CdSe/ZnS-treated groups, respectively. The high percent of mapped reads indicated that the cDNA sequence data accurately corresponded to the transcriptional expression in *S. cerevisiae,* and the multiple sequence alignments indicated the successful alignment of our fragmented sequence data to their homologous segments on the reference genome. The gene identities were also accurately identified.

A total of 7127 genes, including non-coding cDNA, were identified, and of those, 4478 genes were found to have significant changes in transcript expression (*q* < 0.05) when compared to the non-treated controls. From the pool of genes with *q*-values below 0.05, 2267 genes were found to be upregulated, and 2211 genes downregulated. From each pool of up and downregulated genes found to be significantly different, those differentially expressed by a fold of 1.5 or greater were selected (2839 genes). From the gene pool of DEGs with a fold-change of 1.5 and up, we obtained GO terms with GOrilla and found 47.6% (742 of 1560 genes) of upregulated genes involved in cellular nitrogen compound metabolic processes (Figure 2A). Several upregulated genes were implicated in non-coding RNA (ncRNA) processing (18.7%), rRNA processing (14.6%), translation (13.2%), ribonucleoprotein complex biogenesis (13.2%), and cell cycle process (11.6%) (Figure 2A). For the above GO terms, 291 and 228 upregulated genes were involved in ncRNA and rRNA processes, respectively (Figure 2A). Additionally, 206, 206, and 181 genes were implicated in translation, ribonucleoprotein complex biogenesis, and cell cycle processes, respectively (Figure 2A). Enrichment values for GO terms found with genes with a fold-change of 1.5 or greater fell between 1.0 and 2.0, meaning each GO term was approximately as meaningful as any other GO term shown (Appendix A). To gain a clearer understanding of the changes in the cellular transcriptome after treating with CdSe/ZnS-QDs, we selected the 150 most upregulated and 150 most downregulated genes, i.e., 300 genes total, and obtained GO term data consistent with the data represented in Appendix A. We also created a heatmap of these 300 genes to visually depict their highly up- and downregulated expression levels compared to the non-treated controls (Appendix A). We found 102/150 (68%) of these upregulated genes to be involved in cellular component organization or biogenesis, such as ribosomal subunit biogenesis and its assembly to form functional ribosomes. Several of the other highly upregulated genes were involved in ribosomal RNA metabolic processes (63.33%), cleavage involved in rRNA processing (18%), maturation of LSU (large subunit of ribosome, 12.66%) and SSU rRNA (small subunit of ribosome, 18.66%), ncRNA transcription (6%), macromolecule methylation (6%), and genes involved in cell cycle DNA replication (2.66%) (Appendix A). Of the 21 GO terms, generated in Appendix A, five GO terms described the maturation of rRNA, four GO terms were involved in pre-ribosome biogenesis or assembly, four GO terms involved in transcription of rRNA by RNA polymerase I, and three GO terms involved in the transport and export of RNA and ribosomal subunits.

Our GO term analysis on the downregulated genes with a fold difference of at least 1.5 indicated that metabolic processes were negatively affected (Figure 2B). Most downregulated genes were implicated in small molecule metabolic (16.5%) and oxidation-reduction processes (13.0%) (Figure 2B). Several more downregulated genes were involved in carbohydrate metabolic processing (8.0%), responding to chemicals (7.5%), proteolysis (7.5%), ion transmembrane transport (5.7%), import into the cell (3.9%), and the electron transport chain (2.6%) (Figure 2B). For the downregulated GO terms, 211 and 166 genes were implicated in small molecule metabolic and oxidation-reduction processes, respectively (Figure 2B). In addition, 102, 96, 96, 73, 50, and 33 genes were involved in carbohydrate metabolism, response to chemicals, proteolysis, ion transmembrane transport, import into cell, and the electron transport chain, respectively (Figure 2B). Enrichment values for GO terms found with genes with a fold-change of 1.5 or greater fell between 1.0 and 2.0, meaning each GO term was approximately as meaningful as any other GO term shown (Appendix A). From the list of 150 most downregulated genes (Appendix A), we found 26 GO terms on their cellular processes compared to only 21 GO terms pertaining to the pool of 150 most upregulated genes (Appendix A). The downregulated GO terms were found to affect genes involved in various metabolic processes and were more diverse in the cellular processes they effect compared to the upregulated GO terms that are predominantly involved in ribosomal biogenesis (Figure 2A,B, Appendix A). The GO term with the most highly downregulated genes from the list of 150 was the oxidation-reduction process with 31/150 (20.66%) genes involved. Other highly downregulated genes and their GO terms given, based on their cellular processes, were included but not limited to the Generation of precursor metabolites and energy (18%), carbohydrate metabolic processes (16%), cellular response to chemical stimulus (9.33%), alcohol metabolic process (5.33%), antibiotic metabolic process (6%), response to drug (4%), and response to salt stress (3.33%) (Appendix A). Of the 21 GO terms involved in highly downregulated gene pool, 14 were directly involved in metabolism, four GO terms acted as a response to stimuli, such as chemical stimulus, oxidative stress, drug, and salt stress, and three GO terms were found to be directly involved in oxidation processes (Appendix A). We found many GO terms involved in similar processes and several that share many of the same genes and several that do not. For instance, of the four GO terms that represented genes that were involved in responding to stimulus (cellular response to chemical stimulus, response to oxidative stress, response to drug, and response to salt stress) (Appendix A), the gene *CTT1* was involved in each except in the GO term response to chemical stimulus (Appendix A). Interestingly, *NCE103* was found to be involved in the GO terms response to chemical stimulus and oxidative stress, but not in the GO terms response to drug or salt stress (Appendix A). Similarly, *CIN5* was involved in the GO terms response to drug and salt stress but not in the GO terms response to chemical stimulus and oxidative stress (Appendix A). Some GO terms that represent metabolic processes contained the exact same genes, such as the GO terms ethanol metabolic process (four genes involved: *PDC6*, *ALD4*, *ALD6*, and *NDE2*) and alcohol metabolic process (eight genes involved: *DSF1*, *PDC6*, *ALD4*, *GUT2*, *YNR073C*, *YAT2*, *ALD6*, and *NDE2*) (Appendix A). Likewise, there were GO terms that represent metabolic processes that had no genes in common, such as the GO terms polysaccharide metabolic process (10 genes involved: *YMR084W*, *GSY1*, *GLC3*, *GIP2*, *GPH1*, *PGM2*, *GAC1*, *GDB1*, *UGP1*, and *SUC2*) and antibiotic metabolic process (nine genes involved: *CTT1*, *TSA2*, *PDC6*, *ALD4*, *ACH1*, *ALD6*, *NDE2*, *SDH1*, and *SHH4*) (Appendix A).

### 3.3. Validation of RNA-Seq Data by RT-qPCR

We validated our differentially expressed gene data and expression profiles generated from our RNA-seq experiment by conducting a real-time RT-qPCR test. We selected two upregulated genes (*FAF1* and *SDA1*) that play a role in rRNA processing/ribosomal biogenesis, two downregulated genes (*DAN1* and *TIR1*) that form structural mannoproteins that help maintain cell wall integrity, and a housekeeping gene (*ALG9*) whose expression is unchanged in the presence of CdSe/ZnS QDs, based on our RNA-seq fold-change data with a *q*-value less than 0.05. *FAF1* and *SDA1* were found to have 7.32 ± 0.52-fold and 8.06 ± 2.15-fold upregulation in expression, and *DAN1* and *TIR1* were found to have 5.3-fold and 3.3-fold downregulation in expression, respectively, when treated with 10 μg/mL CdSe/ZnS-QDs (Figure 3A,B). The resulting fold-changes for *FAF1*, *SDA1*, *DAN1*, and *TIR1* were measured with RT-qPCR and graphed along with each gene fold-change found with RNA-seq. A linear regression line was drawn to represent the correlation between the fold-changes found with each method, and the RT-qPCR fold-changes were found to be consistent with our RNA-seq expression data (Figure 3C).

### 3.4. ROS in Response to CdSe/ZnS

It is known that cells respond to environmental factors, including nanoparticles, by producing reactive oxygen species that affect the physiology of the cells. To assess the amounts of superoxides produced by the cell when treated with CdSe/ZnS (20 µg/mL) or AgNPs (5 µg/mL), we measured the fluorescent intensities of dihydroethidium (DHE) at 600 nm using a flow cytometer. The rationale for measuring the DHE at 600 nm was to detect the amount of red fluorescence emitted by the oxidation of DHE that is caused by superoxide. The no-cell controls, including only PBS, AgNPs, or CdSe/ZnS-QDs, displayed a little background noise, which was manifested from their low cell counts, along with missing DHE fluorescent intensity peaks (Figure 4A–C). However, the non-treated cell control sample revealed a peak of cells that have oxidized DHE, along with a bigger peak that represents cells carrying non-oxidized DHE. We found that the fluorescent intensities of the oxidized DHE in the AgNP- and CdSe/ZnS-treated cells were not statistically different from that of the non-treated cell control (Figure 4D–G). This suggests that both AgNPs and CdSe/ZnS-QDs have no significant effect on the production of superoxide in the cells.

### 3.5. The Vulnerability of Cell Wall Integrity in Yeast Cells 

Yeast cells maintain a cell wall, which allows the cell to stay well protected from many threats across the board. However, when treated with Zymolase, the cell wall breaks down and leaves the cell with only its plasma membrane, making the cells more sensitive to environmental factors. It is known that nanomaterials are coated with a diverse number of materials, such as sodium citrate coatings on AgNPs and zinc sulfide coatings on CdSe, to minimize their toxicities. Based on the observations made in the previous section (Figure 1), where CdSe/ZnS did not show any significant effects on yeast cell viability, we were interested in observing what effects AgNPs and CdSe/ZnS would have on sensitized yeast cells without cell walls. The no-cell control (Figure 5A) showed that the AgNPs and CdSe/ZnS-QDs induced no change in the optical densities, ensuring that any decrease in optical density during the cell cultures treated with the nanomaterials might be due to cell death with adverse effects by the nanomaterials. The cells compromised by Zymolase showed a modest decrease in cell density over time in the presence of CdSe/ZnS-QDs (20 µg/mL) when compared to the non-treated control, whereas the cells that were not treated with Zymolase showed no significant changes in cell density over time (Figure 5B). Similarly, the AgNPs caused cell density decrease in the presence of Zymolase, at higher concentrations than 5 µg/mL (Figure 5C). The rate of density decrease was more pronounced in the presence of AgNPs than CdSe/ZnS-QDs, and therefore we re-plotted the cell density curves of Zymolase-treated cells from both Figure 5B,C. As a result, we observed that the CdSe/ZnS-treated cells (20 µg/mL) with Zymolase displayed less vulnerability to a decrease in cell density than AgNP-treated cells (5 µg/mL) with Zymolase. This suggests that the cell wall plays a major role in preventing cell death caused by nanomaterials. In this sense, the AgNP-mediated growth defects, shown in Figure 1, appeared to be due to growth delay rather than cell death in the presence of AgNPs. Further, without the presence of the cell wall, both AgNP and CdSe/ZnS-treated cells showed sensitivity to cell death, although in varying degrees.

## 4. Discussion

In August of 2017, the Environmental Protection Agency (EPA) enacted an “Information Gathering Rule”, which requires companies that manufacture or process nanomaterials, regarded as chemical substances, currently in commerce to inform them of the nanomaterials specific chemical identity, production volume, methods of manufacture, processing, use, exposure and release information, and available health and safety data. According to epa.gov, they are attempting to facilitate innovation while ensuring the safety of the nanoscale substances but also states that the information collected on the nanomaterial is not intended to conclude that nanomaterials will cause harmful effects to human health or the environment. The EPA claims that the information gathered is to be used in determining if any further action needs to be taken. In addition, the U.S. Food and Drug Administration (FDA) has established guidelines on assessing the safety, effectiveness, and quality of products containing nanomaterials, and the FDA does not make categorical judgments on the safety or dangers of nanomaterials (epa.gov). Our discussion covers comparisons of old and recent articles, and we hope the data we collected would help expedite the EPA’s decision to take further action. The current investigation contributed to the field of nanomaterial toxicity to gain a better understanding of how CdSe/ZnS-QDs and AgNPs affect living organisms differently and on how nanomaterials of different compositions and shell/core structures interact with cellular environments. To our knowledge, this is the first RNA-seq report on an estimated 4.1 nm CdSe/ZnS-QDs in the budding yeast, *Saccharomyces cerevisiae*, providing a list of differentially expressed genes. Furthermore, we offered a comprehensive model of CdSe/ZnS-QD impacts on cell physiology, which was compared to the previously proposed model that postulates AgNP-mediated changes occurring in yeast.

### 4.1. The Role of the Carboxylic Acid Ligand

Our CdSe/ZnS-QDs were synthesized with a carboxylic acid stabilizing ligand that is capped on the surface on the ZnS shell. The un-dissociated form of a carboxylic acid is lipid-soluble and capable of crossing the membrane by diffusion and can be taken up by specific transport proteins [32]. Once inside the cell, the pH change causes carboxylic acids to dissociate into anions and accumulate because they can no longer diffuse out of the cell. A build-up of protons can increase the acidity of the cytoplasm and change the normal regulation of several metabolic pathways [32]. In addition, a build-up of protons can also generate free radicals that cause oxidative stress. We did not find any increase in ROS, but the accumulation of our QDs in the cell might be altering metabolic gene regulation by decreasing the cellular pH. The budding yeast possesses active transport systems that allow carboxylic acid-containing molecules, such as acetate, pyruvate, and lactate, to cross the plasma membrane [32]. These alternate metabolic pathways are typically turned on in the absence of glucose in a process called the diauxic shift [32]. During the diauxic shift, this study found 700 genes increased in their expression, and 1000 genes decreased in their expression [32]. Interestingly, when comparing yeast cells grown in glucose with cells grown in acetate, genes involved in activating translation machinery, rRNA maturation, and mitochondrial biogenesis were upregulated [32], similar to the results we found in our gene expression analysis. These findings could suggest that many of the upregulated genes found in the presence of our CdSe/ZnS-QDs could result from the carboxylic acid stabilizing ligands capped on to the surface of our QDs.

### 4.2. Why CdSe/ZnS Is Less Toxic Than AgNPs

Of the two nanomaterials, CdSe/ZnS-QDs and AgNPs, the latter was found to have a profound negative effect on cellular proliferation, while no effect was observed in CdSe/ZnS-treated cells (Figure 1). Geisler-Lee et al. (2013) recently demonstrated that approximately more than 10% of AgNPs released Ag^+^ ions in 24 h of exposure in plants [33]. Therefore, we conjectured that the growth defect we observed in cells treated with 20 nm AgNPs was due to, in part, potential leakage of Ag^+^ ions out of the citrate coat. However, the ZnS shell might efficiently prevent the short-term release of Cd^2+^ ions from escaping the core of a Cd-based QD, which led to no growth defects. This assumption can be supported by a previous study that found CdSe/ZnS-QDs, conjugated with COOH, are significantly degraded in cells after two days of exposure. Furthermore, Cd^2+^-mediated toxicity only occurs when cellular Cd^2+^ concentrations exceed a certain threshold, and in highly proliferating cells, in which cell division exceeds the rate of free Cd^2+^ release [3]. Yeast is known to have a 90-min doubling time, and it is likely that its rate of proliferation may exceed the rate of Cd^2+^ release, which could explain the lack of physiological effects seen.

Nonetheless, there appears to be at least 240% more differentially expressed genes (DEGs) in CdSe/ZnS-treated cells than in Ag-treated (Figure 6). This is possibly due to CdSe/ZnS being internalized and trafficked to the nucleus, where it can interact with the biomolecules in the vicinity [34], implicated in particularly with transcription rates. For instance, Cd-QDs can interfere with transcription mechanisms (DNA/RNA polymerases) to alter normal gene expression. To support this claim, a previous publication revealed low levels of Cd^2+^ ions cause significant chromosomal damage in HFF-1 cells exposed to 7.5 nM QDs, while no physiological damage was observed [7]. Amongst the upregulated genes found in CdSe/ZnS- and AgNP-treated cells, we found many similarities in DEGs, such as an increase in rRNA transcription, ribosomal assembly and protein synthesis, tRNA modifications, and nuclear export. Some interesting differences found amongst the statistically upregulated genes between the two nanomaterials is that CdSe/ZnS-treated cells have a drastically higher number of DEGs involved in amino acid metabolic processes. Amongst the downregulated genes found in each treatment of nanomaterial, we found similarities, such as a decrease in cellular ATP production, endocytosis, cell plasma membrane/wall integrity, and responses to oxidative stress [15]. We found a few notable differences amongst the downregulated genes in each treatment, the first being about 10 times more genes involved in responding to chemicals and many more genes that play a role in ubiquitin-mediated late endosome/multivesicular body trafficking/lysosomal degradation in CdSe/ZnS-treated cells.

In both Ag- and CdSe/ZnS-treated cells, we saw no statistical change in the detectable ROS levels. Ting Zhang et al. (2015) measured the levels of four oxidative stress markers, including hydroxyl radicals, in fibroblasts treated with CdSe and CdTe (2.2 nm) QDs, lacking a ZnS shell, at 3.5, 7, and 14 µg/mL and found no difference in hydroxyl radical levels at 3.5 and 7 µg/mL, but indicated a statistical difference in ROS at 14 µg/mL [9]. Therefore, we surmised that CdSe/ZnS-QDs at 10 µg/mL and AgNPs at 5 µg/mL is not a sufficiently high enough concentration to statistically increase the generation of mitochondrial ROS or superoxide levels. Another possible explanation for not detecting a statistical change in mitochondrial ROS is because there are many possible types of ROS produced in the cell, and we selected to quantify only the superoxide for this study. The lack of ROS generation may also be attributed to the presence of the ZnS shell and slower internalization due to the larger diameter [6] of our tested CdSe/ZnS-QDs (4.1 nm).

A previous transcriptomic study on *Chlamydomonas reinhardtii* demonstrated that 20 nm AgNPs (1.5 × 10^5^ mg/L) and 10 nm CdTe/CdS-QDs (2.0 × 10^4^ mg/L) did not induce oxidative stress. The former induced significant damage to the cells’ structural integrity, while green alga cells exposed to CdTe/CdS-QDs did not increase the expression of transcripts that encode proteasome subunits [12]. Consistently, we found several proteasome subunit genes (*RPN1/2/3/4/5/6/7/8/9/11/13/14* and *RPT2/3/4/5/6*) to be significantly downregulated in CdSe/ZnS-treated cells. This similar cellular response suggests that CdTe/CdS and CdSe/ZnS-QD exposure may induce similar transcriptional responses due to their similar composition. Additionally, AgNP-treated cells increased transcript levels that encode for proteins of the cell wall and flagella, suggesting AgNPs are more harmful to structures exposed to the external environment, whereas CdTe/CdS-treated cells downregulated more transcripts overall and resulted in less damage to external structures [12]. Similarly, our Zymolase experiments revealed CdSe/ZnS-QD exposure was less damaging to the cell wall and downregulated more transcripts in yeast than AgNP exposure.

It has long been thought that engineered nanoparticles could affect the integrity of the mechanism of DNA-damage repair pathways, which, in turn, can negatively impact the cellular homeostasis. Our differentially expressed gene list contains genes that function in the yeast base excision repair (BER) pathways, including *APN2, NTG1, NTG2, RAD2, RAD4, RAD5, RAD6, RAD7,* and *RAD9* (data not shown). Ogg1 is also implicated in a BER pathway to excise 8-oxoG from the DNA backbone [36], and we found this gene was highly upregulated. Further, genes implicated in a post-replication uracil excision repair, such as *DUT1, UNG1,* and *REV1* [37], were differentially expressed in the presence of CdSe/ZnS according to our list. Nucleotide excision repair (NER) pathway has the capacity to remove a large number of structurally unrelated helix-distorting lesions [36]. The following genes implicated in the yeast NER pathway were differentially expressed: *ABF1, RAD2, RAD3, RAD7, RAD16*, and *RAD26.* Taken together, our data provides evidence that CdSe/ZnS poses a threat to DNA repair pathways, and therefore, the precise action mechanism behind the threat awaits to be explored.

### 4.3. Upregulated mRNA Transcripts Implicated in Promoting Translation 

We provided a model (Figure 7A,B) postulating potential physiological effects, induced by exposure to CdSe/ZnS-QDs. Our model, therefore, depicted key differentially expressed genes and their corresponding cellular functions. Of these upregulated genes, the most noticeable groups of upregulated genes were for translation, including but not limited to, rRNA transcription, ribosome subunit assembly, ribosome exit, tRNA maturation, and translation machinery assembly in the cytoplasm. Given rRNA synthesis is a prerequisite for translation, our RNA-seq data were consistent in the genes, such as ECM16 [38] and RPA4 [39], required for rRNA synthesis; these genes were highly upregulated (Figure 7A and Appendix A). It is well known that rRNA molecules are associated with pre-ribosomal proteins in the nucleolus to form precursors of large and small ribosomes, pre-large 66S subunit (LSU), and pre-small 40S subunit (SSU), respectively. Our model only provided three genes (*FAF1*, *DBP8*, and *NSA2*) with at least 3-fold increases in their RNA transcripts, among many upregulated genes implicated in pre-ribosome assembly (Figure 7A). The Saccharomyces Genome Database presents the gene products of *FAF1* [40] and *DBP8* [41] that are associated with the assembly of SSU, whereas *Nsa2* functions for LSU assembly [42]. In addition, our RNA-seq revealed that RNA transcripts coding for ribosomal proteins, including *RPS26B*, *RPS3*, and *RPL1B,* were increased by 40–100% (Figure 7A). The rise of these transcripts appears to be necessary to supply the demands for making functional ribosome precursors, such as SSU and LSU, which consist of rRNA and its binding partners, ribosomal proteins. After transportation to the cytoplasm with the aid of nucleoporins, such as Nup2 [43] and Pom152 [44], the SSU and LSU join together along with tRNAs to make a translation-competent supramolecular complex that manufactures proteins de novo to replace nonfunctional proteins that might have been damaged by exposure to ROS [45,46]. In addition to increasing the rate of ribosomal production, we also noticed genes involved in tRNA maturation to be significantly increased to provide additional amino acid products required in translation. Finally, genes that aid in the initiation process of translation, including *FUN12* [47], are upregulated.

### 4.4. Downregulated mRNAs and Their Potential Impacts on the Cell Integrity

Based on our list of GO terms generated with downregulated genes from our RNA-seq data, we illustrated the physiological effects or cellular processes induced by exposure to CdSe/ZnS-QDs (Figure 7B). We found that the exposed cells expressed decreased levels of RNA transcripts involved in oxidation-reduction processes, response to chemicals, pathways of endo/exocytosis, and various metabolic processes (Figure 2). Similar to the model for upregulated genes (Figure 7A), our model for downregulated genes highlighted only a few downregulated genes in the model that represent several more downregulated genes involved in the same cellular process. First, genes involved in endocytosis, including but not limited to *MYO3*, were downregulated. This indicates a potential defect in endocytosis with less number of copies of *Myo3* at the endocytic site, and therefore, it is of great interest in testing whether the endocytic process is hampered in the presence of Cd-QDs. Furthermore, it has been shown that CdSe/ZnS-QDs use endocytosis as the main route for their uptake, according to Liu et al. (2015) [6], but the question of whether they affect the process and rate of endocytosis, directly or indirectly, remains unknown. We found several downregulated genes that play a role in glycolysis. Among these genes, *ENO1* and *GUT1* code for a phosphopyruvate hydratase [48] and a glycerol kinase [49], respectively. Along with these genes, seven *HXT* genes (*HXT 2/17/5/4/9/8/13*) and five *SNF* (*SNF3/1/4/7/11*) genes coding for sugar transporter (SGD) were significantly downregulated based on our RNA-seq in response to Cd-QD exposure. It is highly likely that sugar transport genes and sugar-breaking enzyme genes mentioned above are simultaneously affected by the presence of Cd-QDs. However, we cannot exclude the possibility that *ENO1* and *GUT1* genes are downregulated as a consequence of low levels of sugars transported caused by the suboptimal activity of glucose transporters due to the presence of Cd-QDs.

Other mitochondrial genes (*ATP20, COX7, COX12, COX20, RCF1*, and *QCR6*) involved in respiration aid in ATP synthesis [50], electron transport complexes [51], and cytochrome c oxidase subunits [52,53] were downregulated, suggesting energy production was significantly lessened. These genes code for transmembrane proteins residing at the inner membrane of mitochondria (Figure 7B), and their gene products play a major role in relaying electrons via reduction-oxidation cycles. Additionally, these proteins are aiding in transporting H^+^ ions from the matrix to the inner membrane space to create a proton gradient across the inner membrane. The energetic proton flow down the gradient facilitates ATP formation via the help of ATP20, a part of the ATP-synthase protein complex. Other metabolic genes, *CCP1* and *ALD6*, are required in the citric acid cycle (TCA cycle) and NAD+ regeneration, respectively [54]. Their highly-downregulated expression is suggestive of diminished levels of the electron carriers NADH, available for the electron transport chain (ETC), which may lead to the production of suboptimal amounts of ATP. Interestingly, mitochondrial genes involved in neutralizing ROS, such as *CCP* [54], are downregulated as well, thereby possibly contributing to increased ROS levels that result in cellular damage tied together. Downregulation of genes coding for mitochondrial ETC transmembrane proteins, TCA cycle proteins, and ROS neutralizing proteins might additively or synergistically aggravate mitochondrial functions, which is not a favorable environment to support many cellular processes that require ATP for their action mechanism. However, as depicted in the upregulation model (Figure 7A), we proposed an abnormally elevated translation process upon Cd-QD exposure. This does not seem to be consistent with the diminished level of ATP in cells with Cd-QDs. One possible explanation for this would be that the majority of available energy produced may be directed towards increasing translation. We conjectured that proteins in diverse cellular processes are damaged with the presence of Cd-QDs, making the cell prioritize the replenishment of the damaged proteins.

Late endosome/multivesicular body (MVB) genes (*VPS4/36/55, MVB12, COS1/5/8/10*, and *SHH4*) involved in ubiquitin (Ub)-dependent sorting of receptor proteins for vacuole degradation are significantly downregulated. Ub is a sorting tag that mediates the entry of worn-out receptors into intraluminal vesicles (ILVs) that is targeted to the lysosome or vacuole for degradation [55]. It is well understood how endosomes/MVBs play a role in the balance between recycling and degrading proteins and lipids. This robust balancing act, consequently, contributes to diverse cellular processes, such as nutrient uptake, cell adhesion, cell migration, cytokinesis, cell polarity, and signal transduction [56]. In addition, *UBI4*, a gene that codes for ubiquitin [57], is also found to be downregulated by approximately 100%, according to our data. This suggests that receptor proteins destined for degradation in Cd-QDs exposed cells are not being turned over as efficiently as in healthy cells. With available energy in the cell more limited from mitochondrial damage, the cell likely compensates by downregulating regular processes, such as endosomal sorting and transport pathways. Modifying normal cell functions like the ones mentioned might allow the cells to conserve energy for processes of a higher priority, namely translation, for replacing proteins damaged by CdSe/ZnS-QD exposure. From our DEG analysis, we postulated that worn out and damaged receptor proteins accumulate due to late endosomal and proteasome downregulation. In addition, a previous study on the 20S proteasome subunit in maize revealed that the proteasome plays an important role in providing metal resistance in *Saccharomyces cerevisiae* [58]. These results suggest that the cell is choosing to redirect energy meant for degradation to higher priority processes, while simultaneously compromising its metal resistance. We observed no physiological effects, which suggests this possible energy prioritizing was not great enough to cause significant damage but was still detectable with high throughput sequencing technology.

## 5. Conclusions

The present work provided evidence that CdSe/ZnS-QDs exerted a mild cytotoxic effect on yeast when compared with AgNPs, but it was evident that Cd-QD-treated cells had more differentially expressed genes than AgNPs-treated cells. Our working model behind the steep upregulation in ribosomal biogenesis was most likely due to possible carboxylic acid stabilizing ligands interacting with cellular components or Cd-QD interactions with the damaged proteins as the stable QD particle or as free Cd^2+^ ions released from Cd-QDs. Whereas, a wide spectrum of routine cellular processes, including energy production and intracellular trafficking, appeared to be significantly impeded. We, therefore, proposed that the majority of available energy in the cell is directed to aid translation in order to replenish damaged proteins from Cd-QD exposure.

## Figures and Tables

**Figure 1 biomolecules-09-00653-f001:**
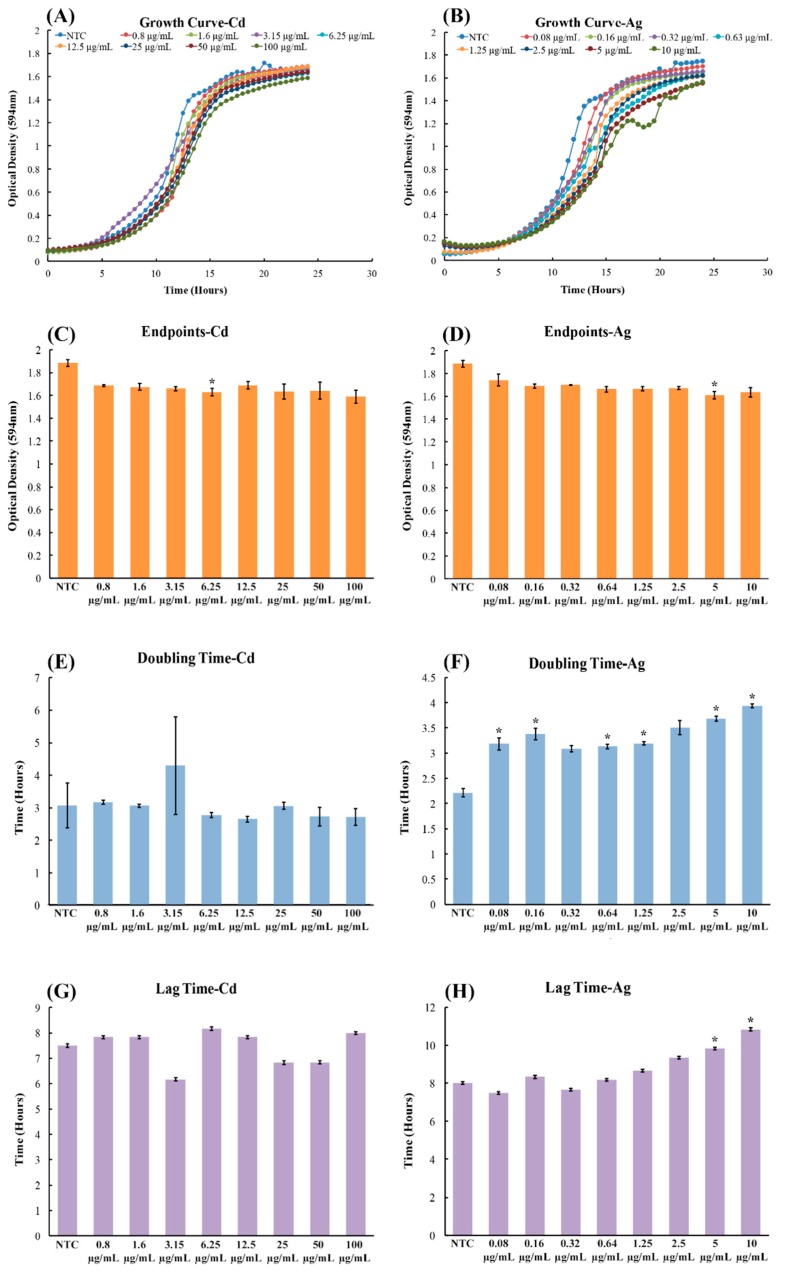
Growth assay to determine growth rates of CdSe/ZnS and AgNP-treated yeast cells. (**A**,**B**) Quantification of cell optical densities over a 24 h period, where the cells were treated with CdSe/ZnS and AgNPs, respectively, at 30 °C while shaking. (**C**,**D**) Measurement of cell optical densities at 24 h of treatment with CdSe/ZnS and AgNPs, respectively. The bar represents the average ODs (600 nm) of each concentration at the 24 h mark. Significant statistical differences are indicated by *p*-values less than 0.05. (**E**,**F**). Doubling time takes place during the phase of exponential growth for the cells treated with CdSe/ZnS and AgNPs, respectively, and was measured as the amount of time it takes for cells to double their ODs. *p*-values of less than 0.05 indicate statistical differences with an asterisk. (**G**,**H**). The mean lag time before the exponential growth phase. AgNP, silver nanoparticle; OD, optical density.

**Figure 2 biomolecules-09-00653-f002:**
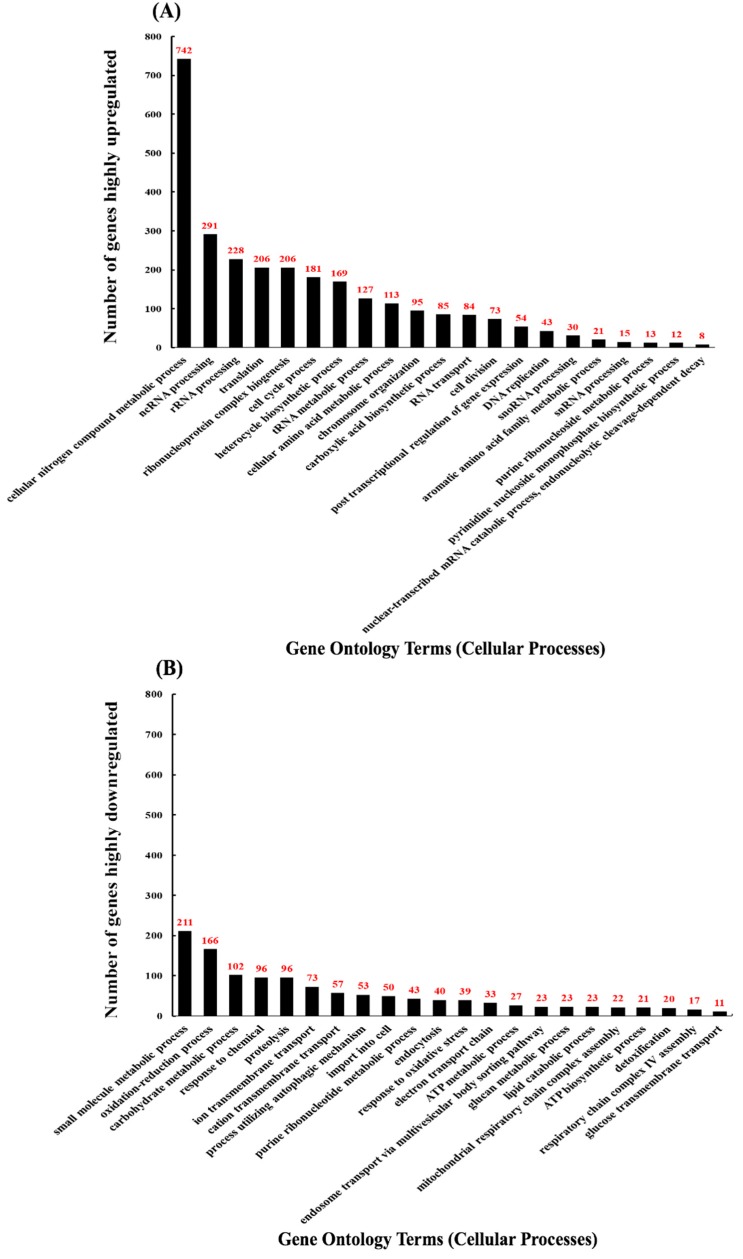
Differentially expressed genes with CdSe/ZnS-QDs (quantum dots). GO (gene ontology) terms corresponding to each differentially expressed gene’s biological process. Out of 4478 genes with a *q*-value below 0.05, 2839 genes with a fold change greater than or equal to 1.5 were incorporated. (**A**) The quantification of upregulated genes associated with their specific GO terms. Of the 2839 statistically different genes, 1560 were found to be upregulated. (**B**) The quantification of downregulated genes associated with their specific GO terms. Of the 2839 genes, 1279 were found to be downregulated.

**Figure 3 biomolecules-09-00653-f003:**
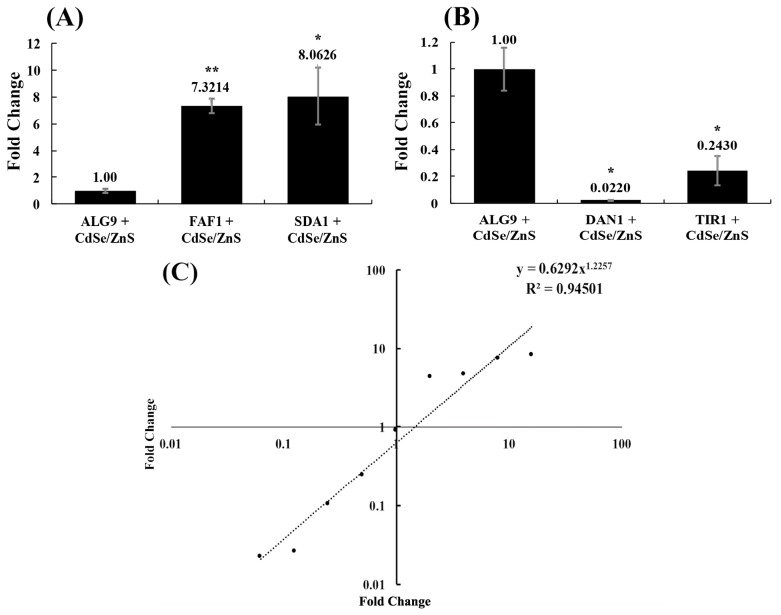
Fold-changes acquired through RT-qPCR. Two up and downregulated genes chosen from our RNA-seq experiments were compared to a housekeeping gene (*ALG9*) found to not be differentially expressed when exposed to CdSe/ZnS-QDs. The fold changes of the two up and downregulated genes were calculated from our RT-qPCR data by utilizing the Pfaffl equation. Fold changes were found with RT-qPCR to validate fold-changes obtained with our RNA-seq. (**A**) The fold changes of the upregulated genes (*FAF1* and *SDA1*) obtained with RT-qPCR. (**B**) The fold-changes of the downregulated genes (*DAN1* and *TIR1*) obtained with RT-qPCR. A student’s *t*-test results are represented with *** (*p <* 0.05), ** (*p* < 0.01). (**C**) The fold change correlation represented by a trend line that shows the power regression line with the equation and R^2^ value of 0.94501. The x and y-axes are in 2- base logarithmic scale, and fold-changes that are <1 and >1 correspond to down and upregulation, respectively.

**Figure 4 biomolecules-09-00653-f004:**
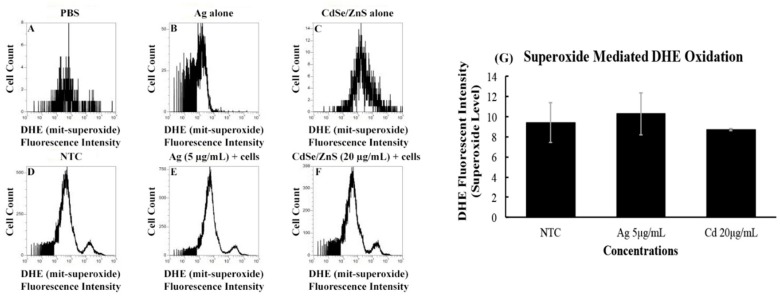
Quantitation of the levels of superoxide produced by cells treated with AgNPs or CdSe/ZnS-QDs. The cells were cultured for six hours with the nanomaterials, then cultured for 2 h with DHE (dihydroethidium) prior to the measurement of the amount of oxidized DHE, which indicates the levels of superoxide produced. (**A**) Non-treated cell control with only PBS and DHE; minimal background noise was detected. (**B**) Non-treated cell control with only AgNPs and DHE in PBS; a slight increase in background noise compared to PBS and DHE alone. (**C**) Non-treated cell control with only CdSe/ZnS and DHE in PBS (the highest background shows a fluorescent intensity detected at 10^3^). (**D**) Non-treated control with cells with DHE in PBS. The major peak indicates the number of cells carrying non-oxidized DHE, while the small peak at fluorescent intensity 10^4.5^ represents the number of cells carrying DHE oxidized by the superoxide produced. (**E**) The effects of AgNPs (5 µg/mL) on the production of superoxide, utilizing a similar method to Figure 4D. (**F**) The effects of CdSe/ZnS (20 µg/mL) on the production of superoxide in cells is indicated by the second peak, as explained in Figure 4. (**G**) Comparison of the percentage of cells that carry oxidized DHE in the non-treated control, AgNPs (5 µg/mL)-treated cells, and CdSe/ZnS (20 µg/mL)-treated cells. Each bar in the graph represents the average of three data sets, and this graph is one representation of three repeated experiments.

**Figure 5 biomolecules-09-00653-f005:**
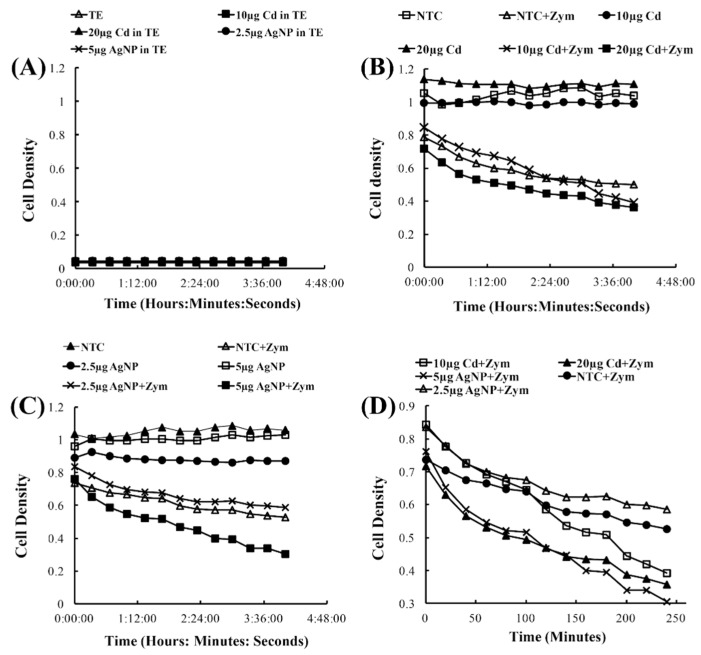
Cell wall viability assay to determine the effects of AgNPs or CdSe/ZnS-QDs on yeast cells lacking cell walls via Zymolase treatment. (**A**) The optical densities of the nanomaterials in TE buffer without yeast cells were measured at a wavelength of 594 nm for four hours while shaking at 30 °C. (**B**) The cell density of yeast cells when treated with differing concentrations of CdSe/ZnS (10 and 20 µg/mL), with or without Zymolase (non-Zymolase-treated cells did not show a significant change in optical density over time). (**C**) Cell density measurements of yeast cells when treated with different concentrations of AgNPs (2.5 and 5 µg/mL), with or without Zymolase. (**D**) Rearrangement of optical densities from Zymolase-treated cells in Figure 5B,C. The non-treated control with Zymolase (filled circle) was compared with 2.5 (empty triangle) and 5 (X symbol) µg/mL AgNP-treated cells, and 10 (empty square) and 20 (filled triangle) µg/mL CdSe/ZnS-treated cells.

**Figure 6 biomolecules-09-00653-f006:**
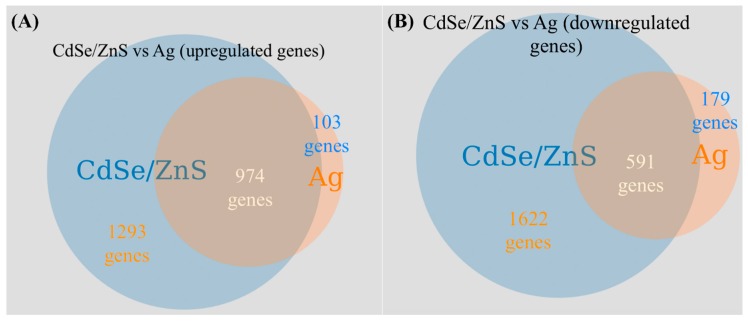
Gene expression Venn-diagram (biovenn.nl) [35] to visualize the shared and separate differentially expressed genes when exposed to CdSe/ZnS-QDs and AgNPs. (**A**) All significant and upregulated genes found in CdSe/ZnS- and Ag-treated cells. The leftmost circle represents the number of genes exposed to CdSe/ZnS, the rightmost circle represents the genes exposed to AgNPs only, and the shared area of the two circles represents the quantity of shared differentially expressed genes in both treatments. (**B**) All significant and downregulated genes. The leftmost circle represents the genes exposed to CdSe/ZnS, the rightmost circle represents the genes exposed to AgNPs, and the middle area represents the same differentially expressed genes in both treatments.

**Figure 7 biomolecules-09-00653-f007:**
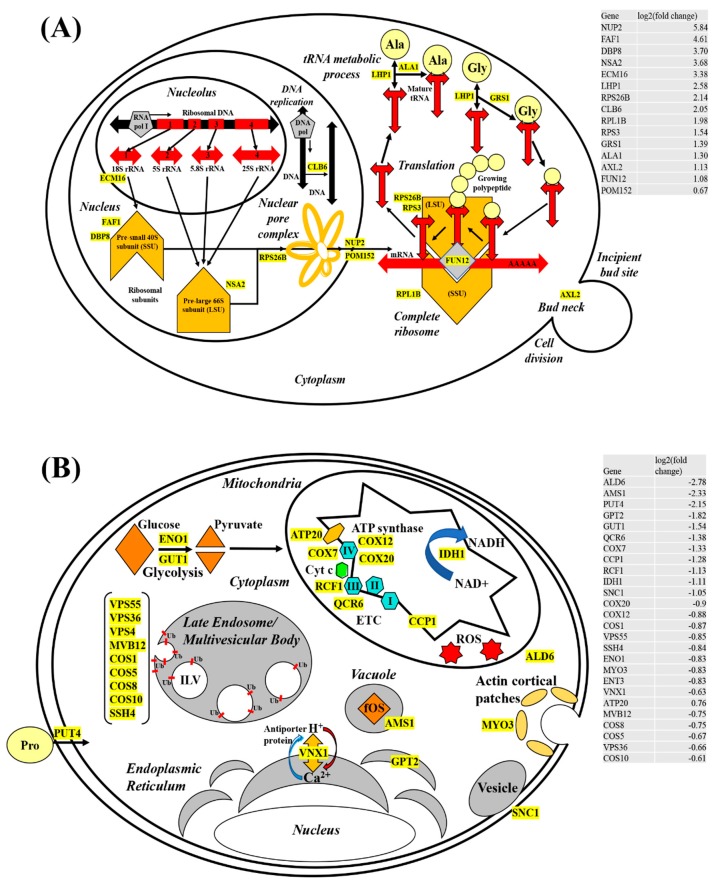
Schematic models of changes in cellular processes with CdSe/ZnS-QDs in yeast cells. Genes are strategically placed near representative illustrations they are thought to be involved in. (**A**) CdSe/ZnS leads to an increase in the expression of genes implicated in the pre-ribosomal assembly of small and large subunits and their nuclear export as well as maturing tRNA and complete ribosomes. (**B**) Several processes appear to be affected by exposure to CdSe/ZnS, including cell wall/membrane integrity, sugar import (see the main text), late endosome/multivesicular body function, and cellular respiration in the mitochondria.

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
