# Peer review of "Transcriptome Profile Alteration with Cadmium Selenide/Zinc Sulfide Quantum Dots in Saccharomyces cerevisiae"

_biomolecules, 2019, doi:10.3390/biom9110653_

Round 1

Reviewer 1 Report

Comments:

Horstmann and co-workers reported an interesting study investigating the QD toxic effects on yeast cells. The methods and biological studies used by the authors are appropriate for such investigation and the data explain main issues well. However, the manuscript would be more effective if some parts are shortened (e.g cDNA sequencing part) providing better clarity for the readers. This is a good first step for understanding the role of core-shell QDs in cells, however their mild cytotoxic effect is concerning in further applications in human cells. I suggest that the manuscript should be accepted after minor revisions noted.

The authors reported that CdSe/ZnS-QD exposure in yeast cells is less damaging to the cell wall and downregulated more transcripts than AgNP exposure. Although they have performed a wide study of transcriptomic analysis, they don’t pinpoint if critical molecules for the DNA damage/repair pathways are affected or not. I think that this is one of the main points of the manuscript and they should explain better. Fig. 3 A and B: y axes need labeling!! In their studies the authors claim that they used CdSe/ZnS QDs with sizes in the range of 4-10 nm, while explaining the role of ZnS shell on preventing the leakage of Cd2+. However, it would be more accurate for their studies to point out the exact size of CdSe/ZnSe QDs used in each case and not only the size-range and the QD concentration. This is a critical point which is also mentioned in the intro, since small differences in terms of QD size have a strong influence on the QD localization.  As an extension to the previous point, have the authors conducted any study relating the effect of QD shell size on cell viability, cell wall stability or gene expression? QDs with thick shells might be more appropriate for their studies. They should discuss this point in the manuscript. Have the authors any plan to perform similar experiments to mammalian cells?

Author Response

Dear reviewers,

I am enclosing here with a revised manuscript Manuscript ID: biomolecules-610782

entitled, “Transcriptome Profile Alteration with Cadmium Selenide/Zinc Sulfide Quantum Dots in Saccharomyces cerevisiae”, for publication in Biomolecules.

Please find the attached “rebuttal letter” in which we provided all necessary explanations that point update sections of manuscript.

If there is any additional information I can provide, please do not hesitate to contact me at kkim@Missouristate.edu

Sincerely,

Kyoungtae Kim

Professor

Department of Biology

Missouri State University

Reviewer 2 Report

Growth assays, RNA-seq, ROS detection assays and cell viability tests were used by the authors to investigate the cytotoxicity of CdSe/ZnS QDs toward yeast cells. The results were compared to Ag(0) NPs used as control. The authors demonstrate that CdSe/ZnS QDs affect genes involved in cellular processes. Some of the results presented are of interest but the manuscript must significantly be improved before acceptance. Here are my comments :

along the whole manuscript, results must be better discussed in the context of literature. introduction : add references related to the toxicity of QDs toward yeast (Small 2012, 8, 2680-2689; J. Lumin. 2018, 194, 760-767; ...). the manuscript suffers from the poor informations given by the authors concerning the nanoparticles used in this work (the sizes of CdSe/ZnS QDs and of Ag NPs must be provided, indicate the structure of the capping ligand for QDs,...). line 99 "brighter emission yield" (?) please clarify. lines 420-421 : the authors cannot compare the citrate ligands used to stabilize and disperse Ag NPs in water to the ZnS shell covering CdSe QDs used to protect the photoactive CdSe core. the stability of QDs and of Ag NPs in the culture medium with or without cells must be evaluated to strenghten the discussion. ICP analyses could be conducted to determine the amounts of Zn2+, Cd2+ or Ag+ leaked by the nanocrystals, if any. the quality of figure 2 must be improved. The text on the x-axis is poorly visible. the length of the manuscript can significantly be reduced. For example, the paragraph "known potential mechanisms of Cd-based QD toxicity" should be moved form the discussion to the introduction. I also suggest to the authors to focus on yeast cells.

Author Response

Dear reviewer 2,

We attached the rebuttal letter pointing out what we have changed in the text.

If you have any questions, please let us know.

Thanks. 

Kyoungtae Kim

Round 2

Reviewer 2 Report

Almost all corrections were made by the authors. The manuscript can be accepted by Biomolecules.